# Comparison of Quantification Methods to Estimate Farm-Level Usage of Antimicrobials in Medicated Feed in Dairy Farms from Québec, Canada

**DOI:** 10.3390/microorganisms9091834

**Published:** 2021-08-30

**Authors:** Hélène Lardé, David Francoz, Jean-Philippe Roy, Marie Archambault, Jonathan Massé, Marie-Ève Paradis, Simon Dufour

**Affiliations:** 1Department of Pathology and Microbiology, Faculty of Veterinary Medicine, Université de Montréal, 3200 rue Sicotte, Saint-Hyacinthe, QC J2S 2M2, Canada; marie.archambault@umontreal.ca (M.A.); jonathan.masse.1@umontreal.ca (J.M.); 2Regroupement FRQNT Op+lait, 3200 rue Sicotte, Saint-Hyacinthe, QC J2S 2M2, Canada; david.francoz@umontreal.ca (D.F.); jean-philippe.roy@umontreal.ca (J.-P.R.); 3Groupe de Recherche en Épidémiologie des Zoonoses et Santé Publique, Faculty of Veterinary Medicine, Université de Montréal, 3200 rue Sicotte, Saint-Hyacinthe, QC J2S 2M2, Canada; 4Ross University School of Veterinary Medicine, P.O. Box 334 Basseterre, St. Kitts, West Indies; 5Department of Clinical Sciences, Faculty of Veterinary Medicine, Université de Montréal, 3200 rue Sicotte, Saint-Hyacinthe, QC J2S 2M2, Canada; 6Association des Médecins Vétérinaires Praticiens du Québec, 1925 rue Girouard Ouest, Saint-Hyacinthe, QC J2S 3A5, Canada; marie-eve.paradis@amvpq.org

**Keywords:** dairy cattle, medicated feed, feed additive, farm-level, monitoring, veterinary prescription, feed mill, antibiotic, ionophore, medically important antimicrobial

## Abstract

Monitoring antimicrobial usage (AMU) in dairy cattle is becoming common in a growing number of countries, with the ultimate goal to improve practices, reduce the development of antimicrobial resistance, and protect human health. However, antimicrobials delivered as feed additives can be missed by some of the quantification methods usually implemented. Our objective was to compare three methods of quantification of in-feed AMU in Québec dairy herds. We recruited 101 dairy producers for one year in the Québec province. Quantities of antimicrobials were calculated by farm from: (1) feed mills invoices (reference method); (2) veterinary prescriptions; and (3) information collected during an in-person interview of each producer. We standardized AMU rates in kilograms per 100 cow-years and compared the reference method to both alternative methods using concordance correlation coefficients and Bland–Altman plots. Antimicrobial usage was well estimated by veterinary prescriptions (concordance correlation coefficient (CCC) = 0.66) or by the approximation using producer’s data (CCC = 0.73) when compared with actual deliveries by feed mills. Users of medically important antimicrobials for human medicine (less than 10% of the farms) were easily identified using veterinary prescriptions. Given that veterinary prescriptions were mostly electronic (90%), this method could be integrated as part of a monitoring system in Québec.

## 1. Introduction

In-feed antimicrobials are given to livestock animals for the prevention or treatment of diseases, and, in some countries, as growth promotors for improvement in feed efficiency and weight gain [1,2]. With the current concern of residues from antimicrobials excreted in the environment and the emergence of antimicrobial resistance related to antimicrobial usage (AMU) in production animals, knowing how antimicrobials are used in animal feed is crucial [3].

In 2018, growth promotion claims were removed in Canada from drug labels [4] for medically important antimicrobials (MIA) as defined by the World Health Organization [5]. Because coccidiostats (ionophores and others) are not MIA, they were not targeted by this ban and can still be used as growth promotors. For the dairy industry in Canada, only three classes of antimicrobials are found in drug premixes authorized for use in feed: (1) tetracyclines (chlortetracycline, oxytetracycline); (2) ionophores (monensin, lasalocid); and (3) other antimicrobials (decoquinate, amprolium) [1]. Only monensin can be added to the diet of lactating dairy cows, however, the other mentioned antimicrobials can be used for non-lactating dairy animals. As of December 2018 in Canada, all MIA for veterinary use are sold by prescription only [4]. In the Québec province, a veterinary prescription is mandatory not only for MIA, but also for any medication (including coccidiostats) to be added in animal feed (chapter P-10, r. 12-Regulation respecting the terms and conditions for the sale of medications [6]). Therefore, a dairy producer has to obtain a prescription for medicated feed from their veterinarian. The veterinary prescription is then sent to a feed mill in charge of preparing and delivering the medicated feed in accordance with the prescription.

Quantification and monitoring of AMU is challenging [7,8]. In 2020, a study by our research team used feed mill invoices to quantify in-feed AMU in dairy cattle [9]. Results showed that ionophores represented the most commonly used antimicrobial class in Québec dairy farms. Because of this, in-feed usage was also the second most commonly reported route after intramammary usage (quantities reported in number of Canadian defined course doses for cattle (DCDbovCA)/100 cow-years). As expected, in-feed usage was the most important when quantities were reported in grams/100 cow-years. Monensin was the most frequently used antimicrobial; other antimicrobials (chlortetracycline, oxytetracycline, neomycin, sulfamethazine, and lasalocid) have been occasionally identified in feed. Recently, the authors compared different quantification methods to report AMU from products other than animal feed, and they identified veterinary invoices as a promising way to implement a surveillance system in Québec dairies [10]. However, because antimicrobials used in animal feed are usually not sold by veterinarians, veterinary invoices could not be used to report properly in-feed AMU.

Different methods of quantification could be used to report on AMU in medicated feed in dairy farms: (i) quantities sold and delivered by feed mills to dairy farms; (ii) quantities prescribed by veterinarians; and (iii) quantities approximated from information collected directly from dairy producers. The objective of the current paper was to compare the AMU rate estimated from veterinary prescriptions for medicated feed or from in-person interviews of dairy producers to the AMU rate calculated from feed mill invoices and deliveries (reference method) at Québec dairy farms. A secondary objective was to estimate the proportion of farms for which electronic veterinary prescription data would be available. We hypothesized that data obtained directly from veterinarians or producers would correlate well with data from feed mills, and that the coverage of the electronic prescriptions (vs. paper) would allow information to be easily gathered for further applications.

This study is part of a larger study on AMU and AMR in Québec dairy farms [9,10,11].

## 2. Materials and Methods

The Health Research Ethics Committee of the Université de Montréal approved this study (project number 16-163-CERES-D) on January 2017. Throughout the article, we followed the STROBE-Vet statement guidelines [12,13] for reporting (Appendix A). Medicated feed was defined as any feed for cattle containing an antimicrobial agent (MIA, ionophores, or other anticoccidial drugs).

### 2.1. Recruitment of Participants

We recruited 101 dairy farms (including two organic farms) for one year (2017–2018) in the province of Québec, Canada. Details of the recruitment process are described elsewhere [9]. Briefly, dairy farms were randomly selected in three dairy regions in the province (Centre-du-Québec, Estrie, and Montérégie). The recruitment represented the actual region to province ratio regarding the number of farms: 19, 15, and 10% for Montérégie, Centre-du-Québec, and Estrie, respectively (stratified random sampling). Dairy farms were excluded if: (1) replacement animals were not raised on the farm or some animals were kept in a barn shared with cattle from another farm; (2) a cessation of activity was planned in the coming 12 months; or (3) the farm was already recruited in a pilot stage for the current project. An in-person interview of each producer was performed between January and March 2018. The producer informed the research team whether they used medicated feed during the study, and if yes, which feed mill(s) delivered medicated feed on the farm, and which animals received the medicated feed. Producers also listed all the veterinary facilities they were working with.

### 2.2. Quantification of AMU in Medicated Feed Using Invoices Issued by Feed Mills

After the end of the project (May 2018), a member of the research team contacted each feed mill listed by the 101 producers by phone to collect all invoices issued during the timeframe of the project (the start and end dates in the project slightly varied from one farm to another).

If several feed mills were identified for a farm, each of them was contacted. The complete label of each medicated feed was collected from feed mills to obtain the precise concentration of active ingredients in the delivered feed. The exact amount of medicated feed sold and delivered in each farm of the project was quantified from invoices (in kilograms or tons of feed). Then, precise antimicrobial quantities (in grams) were calculated by thee farm (using the concentration in the feed). Commercial bags come in a standard weight with a stable concentration of active ingredient, whereas custom-made feed is usually sold in bulk, and the concentration of active ingredient in bulk-delivered feed can slightly vary from one delivery to another. For bulk feed, the associated tag containing the exact concentration of active ingredient was collected for each delivery; if a tag was not available, we assumed that the concentration in the feed was stable during the study year, or from one shipment to another.

Intervals between two deliveries for a farm could be variable. For calculations, we only kept invoices issued between the start and end dates of the farm in the project. We hypothesized that the quantity of medicated feed already present in the farm at the beginning (not quantified because sold by the feed mill before the start date) would be equivalent to the quantity at the end of the project (sold before the end but used later).

### 2.3. Quantification of AMU in Medicated Feed Using Veterinary Prescriptions

All veterinary prescriptions for medicated feed were collected for the 101 farms of the project. Electronic prescriptions were downloaded from the main veterinary billing software programs used by Québec’s dairy practitioners (Vet-Expert and Sysvet, Saint-Hyacinthe, QC, Canada). Moreover, for prescriptions not generated in Vet-Expert or Sysvet (hand-written or electronic), a numeric copy was obtained directly from veterinarians listed by each producer as prescribers of medicated feed for the farm. Information kept for analyses were farm ID, animals to be treated (age, weight, and number of animals), treatment duration, and medicated feed information (type and quantity of feed to be medicated, antimicrobial prescribed in the feed, concentration or incorporation rate, prescription issue date and expiry).

We identified three main ways to prescribe an antimicrobial in feed, and thus, three approaches to deduce the quantity of antimicrobials prescribed: (1) prescription of an annual quantity of medicated feed for the farm; (2) prescription of a rate of incorporation of the antimicrobial in complete feed; or (3) prescription of an antimicrobial dose by head and by day for a given period.

Where an annual quantity of medicated feed was prescribed for the farm, the following equation was used to compute the quantity of antimicrobial prescribed:Antimicrobial quantity (g) = quantity of medicated feed (tons/year) ∗ antimicrobial concentration in the medicated feed (g/ton)(1)

When a rate of incorporation of the antimicrobial in complete feed was prescribed, a daily dry matter intake of 2% (0.02 kg of dry matter per kg bodyweight) was used irrespective of the characteristics of the animals (age, breed, lactating or not, etc.). This percentage (2%) was based on the one used by the European Medicines Agency (Appendix 4 of [14]) and is commonly used to estimate the average consumption of cattle, even if it is known to vary with individual and environmental factors. The following equation was used to compute the quantity of antimicrobial prescribed:Antimicrobial quantity (g) = number of animals ∗ average weight of animals (kg) ∗ treatment duration (days) ∗ daily dry matter intake [0.02] (kg/kg-day) ∗ number of ppm of antimicrobial to be incorporated in the feed on a 100% dry matter basis (mg/kg) /1000(2)

When the quantity of medicated feed was prescribed per animal and per day, the antimicrobial quantity was computed as followed:Antimicrobial quantity (g) = number of animals ∗ quantity of medicated feed per animal and per day (kg/day) ∗ treatment duration (days) ∗ antimicrobial concentration in the medicated feed (mg/kg) /1000(3)

For incomplete veterinary prescriptions, the type of missing information was collected (quantity of medicated feed prescribed, number or average weight of the animals, or antimicrobial concentration in the feed). To compute an antimicrobial quantity, several assumptions were made depending on the type of medicated feed (milk replacer, calf feed, or supplement). When missing, we assumed that the lower weight of a newborn calf was 40 kg (for calculation of an average weight) and that the average weight of a calf (0–6 month-old), a heifer (6–24 month-old), and a cow (from first calving) was 100 kg, 300 kg, and 650 kg, respectively. Those standard weights were chosen because they were already used previously to assign Canadian defined doses in cattle [15]. When the lactation duration was not indicated, a 305-day lactation was used. For milk replacers (incomplete prescriptions), we assumed that 150 g of powder was used to prepare 1 L of solution (according to feeding directions of most milk replacers). We assumed a daily milk replacer intake of 10% (0.1 L per kg bodyweight). This percentage (10%) was based on the daily water intake percentage used by the European Medicines Agency (Appendix 4 of [14]) and is commonly used to estimate the average milk consumption of dairy calves, even if it is known to vary with individual and environmental factors. Using the antimicrobial concentration in the powder, we computed the quantity of medicated powder prescribed as follows:Quantity of medicated powder (kg) = number of calves ∗ average weight of calves (kg) ∗ treatment duration (days) ∗ daily milk replacer intake [0.10] (L/kg-day) ∗ 0.150 (kg of powder per L of solution)(4)

Then, the antimicrobial quantity (in grams) was computed by multiplying the obtained quantity of medicated powder (in kg) by the prescribed antimicrobial concentration in the milk replacer (in mg/kg) and divided by 1000 mg/g. For calf feed, when the quantity prescribed was missing, we assumed that 2 kg of medicated feed was prescribed daily per calf (according to feeding instructions of most feeds: 0.5–3.5 kg for calves 0–6 month-old). Finally, for supplements (incomplete prescriptions), we assumed (for any animals on the prescription) a daily intake of 200 mg of monensin or lasalocid per animal and per day (based on commonly used dosages). The goal was to compute a prescribed quantity each time a prescription was identified, using information from the prescription only. Only the number of animals could not be estimated from the prescription when missing. We identified prescriptions with missing number of animals and used the number of animals provided by the producer during the in-person interview.

Annual prescriptions did not necessarily match the project’s start and end dates. We collected the prescription that overlapped with the study period. For prescriptions that did not change during the study, we calculated the antimicrobial quantity based on the prescription for one year. For prescriptions that were not renewed during the project, we calculated the antimicrobial quantity prorated to the number of days until the end date of the prescription. For new prescriptions generated during the project, we calculated the antimicrobial quantity prorated to the number of days from the start date of the prescription.

### 2.4. Quantification of AMU in Medicated Feed Using Information from the Producer

We developed a method to approximate the quantities of antimicrobials used annually in medicated feed from information available on the farm. The in-person interview allowed us to collect important data from each producer: usage or not of medicated feed on the farm, and if so, antimicrobial(s) present in each feed, group of animals receiving medicated feed, number of heifers reared annually, and number of cows. Then, using the previously published Canadian defined daily dosage for cattle (dddbovCA) of each antimicrobial [15] and the standard weight of each type of animal on a farm, we were able to estimate the quantity of antimicrobials. Specifically, for chlortetracycline, lasalocid, monensin, and oxytetracycline, a daily dosage of 0.66, 0.89, 0.53, and 1.1 mg/kg/day, respectively, was used [15]. For decoquinate, a daily dosage of 0.5 mg/kg/day was used according to the label of Canadian products. Because neomycin sulfate was combined to oxytetracycline at the same concentration in premixes before December 2018, we decided to use the same daily dosage (1.1 mg/kg/day) for both antimicrobials. The same rule was applied to sulfamethazine, which was always combined to chlortetracycline and was assigned a daily dosage of 0.66 mg/kg/day. We only used three standard animal weights for calculations: 100 kg (calves 0–6 months), 300 kg (heifers 6–24 months), and 650 kg (dairy cows >24 months). The period during which the animals were exposed to medicated feed was estimated in days based on the producers’ declaration. The annual quantity of antimicrobials used for each feed was calculated in grams by multiplying the number of animals at risk (number of calves or heifers raised annually, for example) by (1) the number of days medicated feed was distributed (90 days for a 3-month duration, for example), (2) the daily dosage for the antimicrobial used, and (3) the standard weight of animals at risk. 

Quantification of AMU in medicated feed from feed mill sales, from veterinary prescriptions, and from producers’ interviews were reported as methods REF (reference method), VET, and FARM, respectively.

### 2.5. Statistical Analyses

The proportion of veterinary prescriptions that were in an electronic (vs. paper) format was computed along with its 95% confidence interval (95% CI). Then, for each quantification method (REF, VET, and FARM), quantities of antimicrobials (in grams per farm for the study period) were divided by 1000 and standardized by 100 cows to report AMU rates in kilograms of antimicrobials/100 cow-years. The number of cows was obtained from the in-person interview of the 101 producers (total of milking and dry cows at the time of the interview) and assumed to be stable over time.

For each method of quantification, the least square mean AMU rate was estimated by antimicrobial and globally using negative binomial regression models built in SAS software (SAS 9.4 TS Level 1M5, Copyright (c) 2016 by SAS Institute Inc., Cary, NC, USA.), with the quantity of antimicrobials (in kilograms) as outcome, the natural logarithm of the number of cow-years of follow up as offset, and without including any fixed predictors. Intercept estimates and their 95% CI were then back-transformed and multiplied by 100 to report AMU rate and 95% CI in kilograms/100 cow-years. If a problem of dispersion was identified (defined as a Pearson chi2/degrees of freedom <0.80 or >1.20), then robust variance was used to compute 95% CI.

Lin’s concordance correlation coefficients (CCC) were then computed using statistical software R (R version 4.1.1, Copyright (C) 2021 The R Foundation for Statistical Computing, package ‘epiR’ version 2.0.33) to compare the total AMU rate obtained from the REF method to the total AMU rate obtained from the VET or FARM method. A value of 1 for the CCC indicates perfect agreement. Values lower than 1 could indicate a location-shift from the equality line (identical slope but different intercept), a scale-shift (different slope between the one obtained using the observed data and the equality line), or a dispersion of observed data around the observed regression line (i.e., the usual Pearson or Spearman coefficients) [16,17]. The CCC thus takes into consideration, not just clustering of observations around the regression line, but also systematic biases (i.e., difference of absolute values obtained by two methods). We applied the scale developed by Landis and Koch [18] for the kappa statistic to interpret the strength of agreement based on the CCC value obtained: almost perfect (0.81–1.00), substantial (0.61–0.80), moderate (0.41–0.60), fair (0.21–0.40), and slight (0.00–0.20). Bland–Altman diagrams were plotted using R statistical software (package ‘ggplot2’ version 3.3.5) to visualize the 95% limits of agreement and mean differences (biases) between total AMU rates obtained from method REF vs. from method VET or FARM [19,20]. Mean differences were reported with the 95% limits of agreement by antimicrobial, by category (MIA or non-MIA), and in general (package ‘epiR’ version 2.0.33).

For each quantification method, the analyses were repeated using the DCDbovCA unit instead of kilograms. The AMU rates were estimated in DCDbovCA/100 cow-years and the CCC and mean differences (Bland–Altman) were calculated as described for the AMU rates in kilograms/100 cow-years. Results from analyses using the DCDbovCA unit are presented in Appendix B.

## 3. Results

### 3.1. Descriptive Data for the Reference Method REF

Quantification of AMU from feed mill invoices has been previously described [9], except for the usage of decoquinate (not reported previously because not listed by the WHO or Health Canada). A total of 73% of dairy farms (74/101) had at least one medicated feed sold and delivered by one of the 31 feed mills identified during the timeframe of the project (including 13 farms that only used decoquinate in medicated feed). In terms of number of different antimicrobials delivered, 27% (27), 50% (51), and 21% (21) of farms used no antimicrobial, one antimicrobial, and two different antimicrobials in feed, respectively; only two farms used three and five antimicrobials, respectively. Out of 137 different medicated feeds delivered by feed mills to the 74 farms, 133 contained only one antimicrobial: monensin (100 feeds), decoquinate (30), oxytetracycline (two), or lasalocid (one). Two feeds contained two antimicrobials: combined chlortetracycline and monensin (one farm, feed for calves), and combined oxytetracycline and neomycin (one farm, milk replacer). Two feeds contained three antimicrobials: combined lasalocid, chlortetracycline, and sulfamethazine (one farm, supplement for calves), and combined monensin, chlortetracycline, and sulfamethazine (one farm, feed for calves). Among the 74 farms that purchased medicated feed from feed mills, 43% (32), 34% (25), 18% (13), and 5% (four) used 1, 2, 3, and 4 different medicated feeds, respectively. Most delivered medicated feeds were dedicated to calves 0–6 months (102/137; 74%), followed by feeds for cows (lactating or dry) aged 24 months or older (23/137; 17%), and then by feed for heifers 6–24 months (12/137; 9%). Only 4% (6/137) of medicated feed contained at least one MIA, and were delivered by feed mills to 5% (5/101) of the farms (including one farm that used two different feeds containing MIA): milk replacer containing oxytetracycline alone (two farms) or combined with neomycin (one farm), and feed for calves containing chlortetracycline alone (one farm) or combined with sulfamethazine (two farms).

### 3.2. Descriptive Data for the Alternative Method VET

Forty-eight different veterinarians (from 25 veterinary facilities) prescribed at least one medicated feed during the timeframe of the project, for 74% of the farms (75/101). A median of one prescription for medicated feed was obtained by farm (mean of 1.5 prescriptions/farm, range 0–5). Out of 152 veterinary prescriptions, only 10% (15/152) were handwritten, whereas 90% (137/152) were electronic. Among electronic prescriptions, 52% (71/137) originated from Vet-Expert software, 27% (37/137) from Sysvet software, and 21% (29/137) were computer-generated by the veterinarian. Twenty prescriptions (13%; 20/152) could not be associated with any purchase of medicated feed from a feed mill. For 11 of them, no purchase at all could be found, and for nine prescriptions, a direct purchase of the premix from the veterinarian could be associated. For the 11 prescriptions with no purchase at all (nine for monensin, one for decoquinate, and one for lasalocid), the producer stopped using the medicated feed prescribed while the prescription was still in effect during the project. For the nine prescriptions with a purchase directly from the veterinarian, two were for a product containing decoquinate, five for a product containing a combination of chlortetracycline and sulfamethazine, one for a compound containing sulfadiazine and trimethoprim, and one was for a compound containing sulfadiazine, trimethoprim, neomycin, benzylpenicillin, and streptomycin. Five products (5/137; 3.6%) were delivered by feed mills without any veterinary prescription associated: one total mixed ration for calves containing 19.14 mg/kg monensin (only a prescription for complete feed for calves containing 40 mg/kg monensin was identified, but both products were delivered to the farm), one complete feed for calves containing 40 mg/kg monensin (limited purchases; only a prescription for decoquinate was identified for the farm), one complete feed for calves containing 50 mg/kg monensin (limited purchases; only a prescription for decoquinate was identified for the farm), one complete feed for calves containing 50 mg/kg decoquinate (expired prescription for the duration of the project), and one milk replacer containing both 389 mg/kg oxytetracycline and 389 mg/kg neomycin sulfate (we found out that a prescription for another farm outside of the project was used for one of the participating farms). Eight percent (12/152) of prescribed medicated feed contained at least one MIA, and were aimed at 9% (9/101) of the farms (including six farms with one feed prescribed, and three farms with two feeds prescribed): oxytetracycline (two farms), chlortetracycline alone (one farm) or combined with sulfamethazine (seven farms), combined trimethoprim and sulfadiazine (one farm), combined trimethoprim, sulfadiazine, neomycin, benzylpenicillin, and streptomycin (one farm). In six prescriptions, the concentration of the antimicrobial to be introduced in the feed was missing. In 29 prescriptions, the antimicrobial concentration in the medicated feed delivered by feed mills was different from the concentration prescribed by the veterinarian. In 52% of the cases, the error in the concentration prescribed was due to a change in the concentration of monensin and decoquinate in all commercial products of one company in 2017 (the concentration changed in medicated feed for calves from 22.4 to 50 g of monensin per ton, and in milk replacer from 50.16 to 25 g of decoquinate per ton).

Equations (1), (2), (3), and (4) were used 59%, 15%, 25%, and 1% of the time for calculation of the prescribed quantity of antimicrobial. For 18% (30/167) of the antimicrobials prescribed, at least one form of information was missing from the prescription (age or weight of animals, concentration prescribed, or the number of animals), and an assumption had to be made to deduce an antimicrobial quantity. More importantly, for 13% (22/167) of antimicrobials prescribed (14% (22/157) of veterinary prescriptions), the number of animals to be treated was not indicated and this information was instead obtained from the interview with the producer (in order to avoid missing data for comparisons between methods of quantification). Thus, 14% (22/157) of all prescriptions did not contain enough information to directly compute the antimicrobial quantity from prescription data.

### 3.3. Descriptive Data for the Alternative Method FARM

Quantities calculated from the answers of producers to in-person interviews revealed 123 different medicated feeds. A total of 73% (74/101) producers told us that they used antimicrobials in feed. Only 5% (5/101) declared using MIA in feed (same five farms as the one identified by the reference method). Among these five producers, two reported using MIA temporarily for an outbreak of diarrhea and pneumonia in calves (*n* = 1) or for an outbreak of diarrhea in calves (*n* = 1).

### 3.4. Comparisons between Methods of Quantification

One quarter of the farms (25/101) had no invoice from feed mills, no veterinary prescription, and no medicated feed declared in their in-person interview. For 75% of the farms (76/101), at least one quantity was reported by one of the three quantification methods (REF, VET, or FARM). The distribution of total usage rates using methods REF, VET, or FARM was similarly right-skewed, as illustrated in Figure 1A–C (distributions in kg of antimicrobials/100 cow-years) and 1D–1F (distributions in DCDbovCA/100 cow-years).

Proportions of dairy producers using medicated feed, and estimates of usage rates using each of the three methods are presented by antimicrobial, by category (MIA or non-MIA), and in general in Table 1. Regardless of the method used, the most importantly reported antimicrobial in feed was monensin, followed by decoquinate. The REF method estimated a numerically higher total usage rate than the VET and FARM methods, but this finding was not statistically significant.

Agreement between the REF method and VET or FARM method is depicted for the total usage rate in Figure 2A,B (CCC plots), and in Figure 2C,D (Bland–Altman plots). When comparing VET to REF, the regression line was strongly affected by values from 10 farms, for which veterinary prescriptions were observed, but with no (or a substantially lower) corresponding feed mill sale. When refitting the models without these 10 farms (farms with a total AMU rate greater than 2.5 kg/100 cow-years with the VET method and lower than 2.5 kg/100 cow-years with the REF method), a CCC of 0.73 (95% CI: 0.63 to 0.81) was obtained, instead of 0.66 (95%CI: 0.55, 0.76) when no farm was excluded from calculations.

Agreement between methods is presented by antimicrobial, by category (MIA or non-MIA), and in general in Table 2. For the total AMU rate in feed, a substantial agreement was observed between REF and VET methods (CCC = 0.66; CI 95%: 0.55, 0.76), and between REF and FARM methods (CCC = 0.73; CI 95%: 0.63, 0.81). Alternative quantification methods VET and FARM quantified a total usage rate of 0.06 and 0.44 kg antimicrobials/100 cow-years lower than the REF method, but this difference was clinically small and not statistically significant. Between VET and REF methods, a substantial agreement (between 0.61 and 0.80) was identified for monensin, chlortetracycline, and sulfamethazine, a moderate agreement (between 0.41 and 0.60) for decoquinate, and a fair agreement (between 0.21 and 0.40) for lasalocid and oxytetracycline. Between the FARM and REF methods, a substantial agreement was identified for monensin, lasalocid, and neomycin, and a moderate agreement for decoquinate, chlortetracycline, and sulfamethazine. Other antimicrobials only showed a slight (between 0.00 and 0.21) agreement or were not evaluated by the CCC (no user according to one of the two methods compared). For the total of non-MIA, a substantial agreement was found between each alternative method (VET or FARM) and the reference method (REF). For the total of MIA, only a moderate agreement was found between each alternative method (VET or FARM) and the reference method (REF).

## 4. Discussion

To our knowledge, this original study was the first to examine and compare different ways to quantify AMU in medicated feed of dairy cattle. Other authors have reported AMU in dairy cattle feed as a percentage of users, for example, 18.6% of producers used medicated milk replacers in Pennsylvania [21], 25.3% of producers of the United States used monensin in weaned calves on conventional farms (vs. 0% on organic farms) [22], and 4% of cow-calf herds used MIA in feed in Western Canadian cow-calf herds [23]. Special characteristics of medicated feed in comparison to other routes of administration are that (1) a group of animals with common characteristics receives the medication for an extended period that could vary from one herd to another, and (2) the exact dose received per day and per individual is imprecise (the medication is incorporated to the feed for an average individual and based on an average consumption). Calculation of standardized doses for antimicrobials administered in animal feed is therefore a challenge, but has already been achieved to report AMU in dairy cattle in a number of Canadian defined course doses for cattle (DCDbovCA)/100 cow-day [9] and in beef feedlots in a number of animal daily doses (nADD)/100,000 cattle in placement cohort [24]. Results are not easily grasped using standardized doses for medicated feed, and authors in the study on beef feedlots also reported quantities in total grams of antimicrobial drug (gAMD)/100,000 cattle in placement cohort [24].

In the present study, we decided to report quantities primarily in kilograms of antimicrobials per 100 cow-year (the results in DCDbovCA per 100 cow-years are available in Appendix B). Reporting AMU from medicated feed in a net mass of antimicrobials instead of using defined doses should ease comparisons with other studies as no consensus currently exists on how defined doses are assigned for antimicrobials found in premixes in dairy cattle. Converting a mass (in kilograms) of antimicrobials into a number of defined daily (or course) doses is easily performed by dividing the mass by the chosen defined dose. An average of 2.5 kg antimicrobials or 132 DCDbovCA (method REF) was used annually in feed in a standard 100-cow Québec dairy farm. When using the unit “mg/PCU” developed by the European Medicines Agency (with the standard weight of 425 kg for a living dairy cow) [25], we found a median of 10.2 mg/PCU (REF method), 15.5 mg/PCU (VET method), or 4.4 mg/PCU (FARM method). From previous data on the same sample of 101 dairy farms [9], we calculated a median of 29.7 mg/PCU for the overall AMU of other routes of administration (all routes apart from in-feed usage). Combined results (mg/PCU from in-feed usage and other routes of administration) for Québec dairy farms were similar to the findings of a larger study carried out in 2017 in 41 countries [26]. In this study, authors collected antimicrobial sales data for chicken, cattle, and pig systems and identified that pigs used 193 mg/PCU, chickens 68 g/PCU, and cattle 42 mg/PCU of antimicrobials on average.

We showed that quantification of AMU in medicated feed was feasible using veterinary prescriptions and found a substantial agreement with the quantification using feed mill invoices. Veterinary prescriptions identified a higher proportion of farms using MIA in feed in comparison with feed mill invoices (9% vs. 5%). The reason could be that MIA were prescribed and sold at the same time by the veterinarian to treat an outbreak of diseases, temporarily. However, the proportion of dairy farms using MIA in feed was low regardless of the method used, and questioned the necessity of surveillance of medicated feed in Québec dairy farms.

Collecting data directly from dairy producers showed a substantial agreement with feed mill invoices. The type of medicated feed used on a given farm seemed to be quite stable over time, which could facilitate the collection of information (once a year, for example). Only a few producers (7/101) changed from one feed to another during the timeframe of the study: cessation of the usage of antimicrobials for the benefit of essential oils and spice extracts as feed additives for lactating cows, or shift from decoquinate to monensin (or inversely) for calves, as examples. Alternatives to antimicrobials as feed additives in dairy cattle is increasingly sought, and essential oils are promising candidates: some suggested improvement in feed efficiency after weaning compared to monensin [27], or a positive impact on milk yield, milk composition, and feed efficiency in lactating cows [28].

Both reference and alternative quantification methods had limitations. Regarding invoices obtained from feed mills, information can be imprecise, especially for bulk feed that was delivered six to 12 times a year. Feed mills consist of numerous private companies in Québec, and owners were sometimes reluctant to share the incorporation rate of active ingredients in custom-made feed (or did not always keep track of previous preparations; we had to assume that the incorporation rate did not drastically change from one preparation to another). Collecting invoices from feed mills was a long and tedious process that depended on the transparency and willingness of millers to share data. In addition, the number of invoices could be important to analyze for some farms (one invoice every two weeks, containing not only information on medicated feed), and data were not homogenous from one feed mill to another (different software and ways of invoicing).

Regarding veterinary prescriptions, collecting data was easily performed (90% of data being electronic, and veterinarians were inclined to share information), but time-consuming, especially for prescriptions not generated in Vet-Expert or Sysvet. Missing data in the prescriptions were the main limitations of the method (for 14% of prescriptions, the estimation of the quantity prescribed could not have been feasible without collecting information on the number of animals in the farm, directly from the producer). Some information was also incorrectly completed in prescriptions. The number of animals was sometimes recorded as the total number of calves or heifers raised on the farm per year or the average number of calves or heifers at a given time on the farm (for example, 30 or three animals on the prescription, for the same size of herd). The duration of treatment on the prescription was often mistaken for the period of validity of the prescription (duration of 365 days instead of four months, for example, if prescribed for calves aged two to six months). Veterinarians were rarely identified as sellers of premixes. This information was deemed important because without any conflicts of interest, veterinarians could be more willing to share their prescriptions (unbundling of prescription and sale of antimicrobials for medicated feed). Our results showed that prescriptions could miss 3.6% of medicated feed actually delivered to farms: in some cases, the prescription had expired or the medicated feed delivered was a small quantity (maybe for animals other than dairy cattle on the farm). One product only raised questioning (milk replacer containing MIA found on feed mills invoices for a given farm, with a prescription identified for a neighboring farm). This practice is illegal, but was an isolated occurrence in the project.

In Québec province specifically, because most veterinary practitioners use the same software for their accounting (Vet-Expert mainly and, to a lesser extent, Sysvet), most veterinary prescriptions were expected to be in an electronic, standardized format, and, for Vet-Expert software, centralized in a server. This database could, therefore, be used for ‘one-stop’ shopping of data on AMU in medicated feed in dairy farms, in addition to the other more common administration routes (already described in [10]).

Finally, gathering annual answers to surveys from every producer of the province may be a valuable way to obtain data on in-feed AMU in dairies. The method we used to estimate quantities of in-feed antimicrobials from questioning every producer of the project was based on approximations (weight of animals, average daily dosage for each antimicrobial, average number of animals raised on the farm annually) and did not allow us to obtain precise quantities, but correlated well with the reference method. The FARM method was deemed good for periodic quantification of in-feed AMU.

In our study, antimicrobial agents identified in medicated feed were mainly monensin and decoquinate (59% and 28% of users, respectively). In comparison, a similar proportion of users was found in Australian feedlots for monensin (61% of respondents), but higher proportions were identified for MIA (20%, 4%, and 15% of users for virginiamycin, tylosin, and tetracyclines) [29]. In a study on 36 beef feedlots in Western Canada, 97% of cattle were exposed to MIA in feed (mainly chlortetracycline) [23]. In intensive livestock operations, cattle have a higher infection pressure (higher density of animals from different origin, grain-based diet, rapid growth), and this could explain the preventive use of antimicrobials (prevention of liver abscesses, respiratory diseases, pink eye, foot rot, etc.). In a study on American conventional dairy farms (2000–2001 data), 17% reported using MIA in the feed, but only 4% continuously [22].

Ionophores are a class of compounds regarded as feed additive, Category IV (antimicrobial agents of low importance not used in human medicine) in Canada [30], and non-MIA [5], that exhibit both antimicrobial and anticoccidial activity. A recent study showed that monensin could be concentrated in surface water adjacent to cattle grazing areas [31]. Cross-resistance between different ionophores was reported [32], but cross-resistance between ionophores and other classes of antimicrobials were infrequently identified. However, recent findings showed that narasin and vancomycin resistance could be physically linked on transferrable plasmids in *Enterococcus faecium* [33]. A study also suggests that some ionophores (narasin, maduramicin, and salinomycin) could contribute to the persistence of vancomycin resistant *Enterococcus faecium* in poultry livestock [34]. However, research on mechanisms of resistance to ionophores, and co-selection of resistance for MIA is still in its infancy. A study on the bovine gut microbiome and antibiotic resistome did not show any correlation between the presence of antimicrobial resistance genes in the gut microbiota and the use of monensin and tylosin as feed additives [35]. Because ionophores are not used in human medicine, and co-selection of resistance is still infrequent and not completely understood, ionophores are currently of little concern for human medicine.

## 5. Conclusions

A substantial agreement was found between the method using veterinary prescriptions and the reference method using feed mill deliveries to quantify in-feed AMU in Québec dairy farms. A similar agreement was calculated between the method using answers of producers to an in-person interview and demographic data (herd size, number of cattle per age group) and the reference method. These key findings are essential to plan the development of a monitoring system, as existing systems frequently miss data on in-feed usage. In Québec province, using veterinary invoices, for example, without including data on prescriptions for medicated feed, could led to an important underestimation of the quantities used in dairy farms through animal feed.

We showed that coccidiostats are the most commonly used antimicrobials in feed. The current usage of MIA in feed on Québec dairy farms appears to be extremely low, and was never identified for lactating or dry cows. Only monensin was identified as a feed additive for lactating cows. Thus, improving monitoring of antimicrobials in medicated feed will probably make little or no difference when estimating only the general usage of MIA in a population of dairy farms.

## Figures and Tables

**Figure 1 microorganisms-09-01834-f001:**
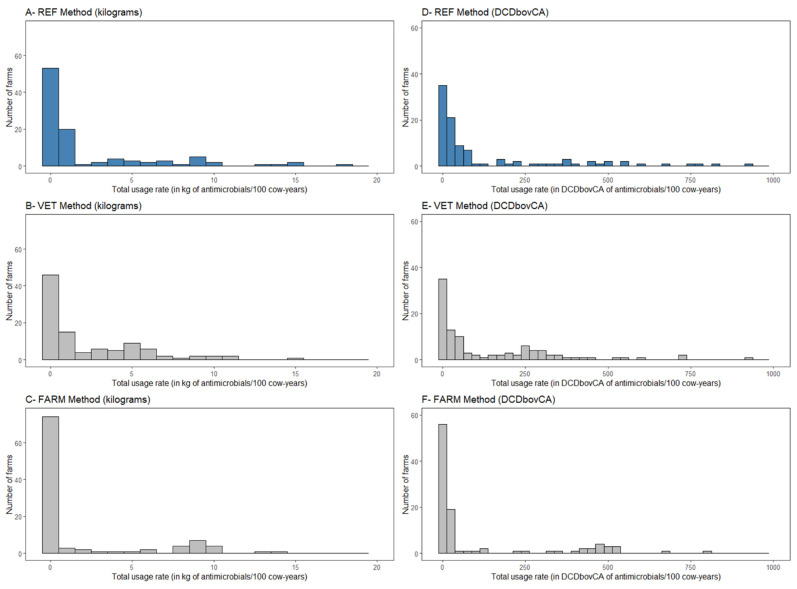
Distribution of total usage rates of antimicrobials in feed (in kg/100 cow-years, (**A**–**C**), and in DCDbovCA/100 cow-years, (**D**–**F**)) calculated for 101 dairy farms from Québec, Canada, using three quantification methods: invoices from feed mills (REF method, histograms (**A**,**D**)), veterinary prescriptions (VET method, histograms (**B**,**E**)), and in-person interviews of dairy producers (FARM method, histograms (**C**,**F**)).

**Figure 2 microorganisms-09-01834-f002:**
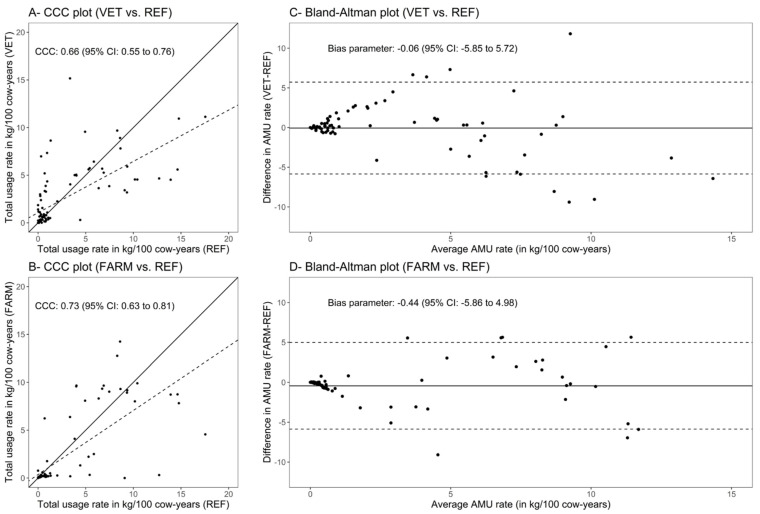
Concordance correlation coefficient (CCC) plots showing strength of association for quantification of total usage rate of antimicrobials (in kilograms /100 cow-years) between the reference method REF and methods VET (Plot **A**) or FARM (Plot **B**). The solid line represents the line of perfect concordance; the dashed line represents the reduced major axis. Bland–Altman plots showing agreement in quantification of total usage rate of antimicrobials (in kilograms/100 cow-years) between the reference method REF and methods VET (Plot **C**) or FARM (Plot **D**). The solid line represents the mean difference (estimated bias); dashed lines represent the 95% limits of agreement.

**Table 1 microorganisms-09-01834-t001:** Proportion of farms (with 95% confidence interval, 95% CI) using antimicrobials in feed, and estimates of usage rate in kilograms per 100 cow-years (with 95% CI) by antimicrobial, by category (medically important antimicrobials, MIA, or non-MIA) and totally, estimated using negative binomial regression models applied to three quantification methods in 101 dairy farms from Québec, Canada: invoices from feed mills (REF method), veterinary prescriptions (VET method), and in-person interviews of dairy producers (FARM method).

Antimicrobial Agent(s)	REF Method		VET Method		FARM Method	
	% of Users (95% CI)	Estimated Rate (95% CI)	% of Users (95% CI)	Estimated Rate (95% CI)	% of Users (95% CI)	Estimated Rate (95% CI)
Monensin	59 (49, 69)	2.24 (1.56, 3.20)	58 (48, 68)	1.90 (1.43, 2.52)	56 (46, 66)	1.89 (1.30, 2.73)
Lasalocid	2 (0, 7)	0.04 (0.01, 0.22)	3 (1, 8)	0.13 (0.03, 0.50)	2 (0, 7)	0.10 (0.01, 0.65)
Decoquinate	28 (19, 38)	0.13 (0.08, 0.19)	29 (20, 39)	0.20 (0.11, 0.34)	27 (18, 36)	0.04 (0.03, 0.06)
Chlortetracycline ^1^	3 (1, 8)	0.04 (0.01, 0.14)	7 (3, 14)	0.07 (0.02, 0.20)	3 (1, 8)	0.01 (0.00, 0.03)
Oxytetracycline ^1^	3 (1, 8)	0.02 (0.01, 0.07)	2 (0, 7)	0.08 (0.02, 0.35)	3 (1, 8)	0.00 (0.00, 0.01)
Neomycin sulfate ^1^	1 (0, 5)	0.00 (0.00, 0.02)	1 (0, 5)	0.02 (0.00, 0.12)	1 (0, 5)	0.00 (0.00, 0.02)
Streptomycin sulfate ^1^	0 (0, 4)	0.00 (NA)	1 (0, 5)	0.00 (0.00, 0.00)	0 (0, 4)	0.00 (NA)
Benzylpenicillin ^1^	0 (0, 4)	0.00 (NA)	1 (0, 5)	0.00 (0.00, 0.00)	0 (0, 4)	0.00 (NA)
Sulfadiazine ^1^	0 (0, 4)	0.00 (NA)	1 (0, 5)	0.01 (0.00, 0.07)	0 (0, 4)	0.00 (NA)
Sulfamethazine ^1^	2 (0, 7)	0.03 (0.01, 0.15)	7 (3, 14)	0.07 (0.02, 0.20)	2 (0, 7)	0.01 (0.00, 0.03)
Trimethoprim ^1^	0 (0, 4)	0.00 (NA)	1 (0, 5)	0.00 (0.00, 0.01)	0 (0, 4)	0.00 (NA)
Total, non-MIA	73 (64, 82)	2.42 (1.78, 3.30)	74 (65, 82)	2.25 (1.73, 2.92)	72 (62, 81)	2.04 (1.42, 2.94)
Total, MIA	5 (2, 11)	0.09 (0.02, 0.39)	9 (4, 16)	0.22 (0.10, 0.49)	5 (2, 11)	0.04 (0.01, 0.16)
Total	73 (64, 82)	2.50 (1.84, 3.39)	74 (65, 82)	2.46 (1.90, 3.18)	73 (64, 82)	2.06 (1.44, 2.96)

^1^ Medically important antimicrobials (MIA) according to the World Health Organization [5]. NA: not applicable.

**Table 2 microorganisms-09-01834-t002:** Concordance correlation coefficients (CCC) and their 95% confidence interval (95% CI), and mean differences and their 95% limits of agreement, to report agreement for quantification of usage rate of antimicrobials in feed (in kilograms/100 cow-years) between quantification using feed mill invoices (reference method REF) and quantification using veterinary prescriptions (VET method) or in-person interviews of dairy producers (FARM method) by antimicrobial, category (medically important antimicrobials, MIA, or non-MIA), and totally from a sample of 101 dairy farms from Québec, Canada.

Antimicrobial Agent(s)	REF vs. VET		REF vs. FARM	
	CCC (95% CI)	Mean Difference (95% Limits of Agreement)	CCC (95% CI)	Mean Difference (95% Limits of Agreement)
Monensin	0.68 (0.57, 0.76)	−0.36 (−5.69, 4.98)	0.72 (0.61, 0.80)	−0.37 (−5.86, 5.13)
Lasalocid	0.37 (0.27, 0.47)	0.09 (−1.39, 1.57)	0.66 (0.65, 0.67)	0.06 (−1.08, 1.20)
Decoquinate	0.41 (0.29, 0.52)	0.07 (−0.91, 1.05)	0.43 (0.38, 0.49)	−0.08 (−0.50, 0.33)
Chlortetracycline ^1^	0.73 (0.63, 0.81)	0.03 (−0.30, 0.35)	0.54 (0.50, 0.57)	−0.01 (−0.29, 0.26)
Oxytetracycline ^1^	0.38 (0.31, 0.44)	0.06 (−0.90, 1.02)	0.20 (0.17, 0.22)	−0.02 (−0.31, 0.27)
Neomycin sulfate ^1^	0.00 (−0.08, 0.07)	0.01 (−0.28, 0.30)	0.70 (0.69, 0.71)	0.00 (−0.04, 0.03)
Streptomycin sulfate ^1^	^2^	0.00 (−0.01, 0.01)	^2^	0.00 (0.00, 0.00)
Benzylpenicillin ^1^	^2^	0.00 (0.00, 0.00)	^2^	0.00 (0.00, 0.00)
Sulfadiazine ^1^	^2^	0.01 (−0.14, 0.16)	^2^	0.01 (−0.14, 0.16)
Sulfamethazine ^1^	0.73 (0.62, 0.80)	0.03 (−0.28, 0.33)	0.52 (0.51, 0.54)	0.00 (0.00, 0.00)
Trimethoprim ^1^	^2^	0.00 (−0.02, 0.03)	^2^	0.00 (0.00, 0.00)
Total, non-MIA	0.63 (0.50, 0.73)	−0.19 (−6.07, 5.68)	0.71 (0.59, 0.79)	−0.39 (−6.01, 5.23)
Total, MIA	0.51 (0.38, 0.62)	0.13 (−1.13, 1.40)	0.51 (0.48, 0.54)	−0.05 (−0.73, 0.63)
Total	0.66 (0.55, 0.76)	−0.06 (−5.85, 5.72)	0.73 (0.63, 0.81)	−0.44 (−5.86, 4.98)

^1^ Medically important antimicrobials (MIA) according to the World Health Organization [5]. ^2^ Antimicrobials for which both methods perfectly agreed and reported a usage of 0 kg/100 cow-years for all farms. In this case, a CCC could not be computed.

## Data Availability

Restrictions apply to the availability of these data. Some information (veterinary prescriptions) was obtained from third party partners and may be available from the author with the permission of these partners.

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
