# Peer review of "Comparison of Quantification Methods to Estimate Farm-Level Usage of Antimicrobials in Medicated Feed in Dairy Farms from Québec, Canada"

_microorganisms, 2021, doi:10.3390/microorganisms9091834_

Round 1
Reviewer 1 Report
This is a very interesting manuscript and, in addition, it is very well written, so I have not specific remarks, but I would like to pose a couple of questions to the authors.
Lines 336-338. Did the farmers know that the same information will be also obtained from other sources?
Lines 435-437. Do you think that is MIA use underestimated?
Author Response
Comments and suggestions for authors: This is a very interesting manuscript and, in addition, it is very well written, so I have not specific remarks, but I would like to pose a couple of questions to the authors.
Lines 336-338. Did the farmers know that the same information will be also obtained from other sources?
Response 1: The farmers knew that we would communicate with their feed mill(s) and veterinarian(s) to collect antimicrobial information from invoices and prescriptions. The farmers were our primary contact, and they provided this information at the beginning of the project. Because of that, if they did not mention (intentionally or not!) one feed mill or one veterinarian, we were not able to get this information from another source (potential missing data). We believe that this was unlikely as all producers were willing to participate in the project and were very inclined to share their information. What could happen though was a producer that changed from one feed mill to another during the project and forgot to mention this change to the research team. Usually, we noted that the delivery of medicated feed stopped with the feed mill in question. When we contacted the feed mill, the person in charge would mention that they stopped working with the farm and gave us the information of the new feed mill for this farm.
Lines 435-437. Do you think that is MIA use underestimated?
Response 2: We do not think that MIA use was underestimated because we obtained almost the same results from the three quantification methods in regards to the MIA use. Method “VET” identified the higher proportion of MIA users and the higher MIA usage rate because some medicated premixes were prescribed and sold by the veterinarian at the same time (no preparation of the medicated feed by the feed mill in this situation). This situation was rare and usually temporary, for example to treat an episode of diarrhea or pneumonia in calves (not necessarily the right way to solve the problem! But this is what we identified on rare occasions).
Reviewer 2 Report
General comments:
The manuscript was well written in general, however the reviewer still needs to review the other related studies to fully understand the methods. Beside this study, the authors also compared the methods of quantification the usage of antimicrobials at farm-level in other than medicated feed using different AMU rate: the number of Canadian Defined Course Doses for cattle (DCDbovCA) per 100 cow-years. It is unclear why the authors used two different metrics to standardized the AMU rate in two studies that come from the same data sources. As showed in the previous published paper from the authors that compare AMU that did not use in feed (used DCDbovCA per 100 cow-years), the choices of metrics used in each study can significantly affect the finding. Therefore, the reviewer strongly believed that the authors should include the metric: ‘the number of Canadian Defined Course Doses for cattle (DCDbovCA) per 100 cow-years’ in this study. The reason is these two metrics will provide full picture of AMU: the amount of use (kilograms per 100 cow-years) and the frequency of use (number of DCDbovCA per 100 cow-years). Another limitation of this study is that the metric used in this study (and other published studies from same parent project) was unique for dairy industry in Canada (mg or course dose per animal). As a result, comparing with other studies that used ‘mg/kg’ metric which is very common in poultry and swine industry is not possible (ESVAC and FAO report). Additional metric in mg/kg were recommended in order to compare the result with wider international studies and with different animal species.
Limitation and strengths
Strengths:
This study was a part of parent study which is well designed and had high quality of data collection. The manuscript was well written with clear and concise content.
Limitation:
The author made decision that presented only 1 metrics of AMU measurement amount of use (can be considered as quantitative metrics), and missing the frequency of AMU measurement (can be considered as qualitative metrics). The quantitative metrics highly affect by the daily dose of antimicrobials. For example, treatment of Tilmicosin required 175 mg per kg per course, while treatment of Tylosin only require a 19.8 mg per kg per course. If two cows received one of this 5 courses per year, they will receive same frequency of treatment but different in term of amount. Unless one of the three methods used in this study only reported the “amount” information, both metrics should be always presented.
Another limitation of this study is that the metric used in this study (and other published studies from same parent project) was unique for dairy industry in Canada (mg or course dose per animal). As a result, comparing with other studies that used ‘mg/kg’ metric which is very common in poultry and swine industry is not possible (ESVAC and FAO report).
Specific comments:
Line 67-68: please provide more detail explanation why authors choose the metric: ‘grams/100 cow-years’ in this study.
Line 249-250: in medicine CCC value <0.7 is considered as moderate correlation only. Please consider to revise this level of agreement.
Figure 2: please add CCC value and p-value in the plot
Table 2: It seems that all of the CCC was not significant, please add one more column for p-value. Please clearly mention whether the correlation of those AAI were statistically significant or not? It is clearly that except for monensin and decoquinate, the percentage of user of other antimicrobial agents were very low (<3%), calculation of CCC for those AAI were not that helpful because it is not enough data point to obtain statistically significant.
Discussion
Authors discuss thoroughly the strength and weakness of the study and compare the finding with other published studies. However, due to the choice of metrics, the authors were unable to compare the result with other animal species: AMU in medicated feed in swine and poultry, for example. There are several publications related to AMU in medicated feed in swine and poultry recently.
Author Response
Comments and suggestions for authors.
General comments:
The manuscript was well written in general, however the reviewer still needs to review the other related studies to fully understand the methods. Beside this study, the authors also compared the methods of quantification the usage of antimicrobials at farm-level in other than medicated feed using different AMU rate: the number of Canadian Defined Course Doses for cattle (DCDbovCA) per 100 cow-years. It is unclear why the authors used two different metrics to standardized the AMU rate in two studies that come from the same data sources. As showed in the previous published paper from the authors that compare AMU that did not use in feed (used DCDbovCA per 100 cow-years), the choices of metrics used in each study can significantly affect the finding. Therefore, the reviewer strongly believed that the authors should include the metric: ‘the number of Canadian Defined Course Doses for cattle (DCDbovCA) per 100 cow-years’ in this study. The reason is these two metrics will provide full picture of AMU: the amount of use (kilograms per 100 cow-years) and the frequency of use (number of DCDbovCA per 100 cow-years). Another limitation of this study is that the metric used in this study (and other published studies from same parent project) was unique for dairy industry in Canada (mg or course dose per animal). As a result, comparing with other studies that used ‘mg/kg’ metric which is very common in poultry and swine industry is not possible (ESVAC and FAO report). Additional metric in mg/kg were recommended in order to compare the result with wider international studies and with different animal species.
Response: Thank you for your comment concerning the indicators; we agree that the multitude of metrics does not facilitate the comparison between studies and we added the results in DCDbovCA/100 cow-years in Appendix A for a better understanding. (Lines 582 and following). The main objective of the present article was to compare methods of quantification for antimicrobials used in feed, not to report the amount of antimicrobials used in feed. An article on antimicrobial usage quantification, using different metrics, was previously published by our team (Lardé H, Dufour S, Archambault M, Massé J, Roy J-P and Francoz D (2021) An observational cohort study on antimicrobial usage on dairy farms in Quebec, Canada. J Dairy Sci. 2021 Feb;104(2):1864-1880. doi: 10.3168/jds.2020-18848). However, we agree that our study would have a greater reach with results presented using units. We added a paragraph in discussion where we present some results using the mg/PCU metric in order to compare with other studies (lines 430-448) and an Appendix A with results using the unit DCDbovCA.
Limitation and strengths
Strengths:
This study was a part of parent study which is well designed and had high quality of data collection. The manuscript was well written with clear and concise content.
Limitation:
The author made decision that presented only 1 metrics of AMU measurement amount of use (can be considered as quantitative metrics), and missing the frequency of AMU measurement (can be considered as qualitative metrics). The quantitative metrics highly affect by the daily dose of antimicrobials. For example, treatment of Tilmicosin required 175 mg per kg per course, while treatment of Tylosin only require a 19.8 mg per kg per course. If two cows received one of this 5 courses per year, they will receive same frequency of treatment but different in term of amount. Unless one of the three methods used in this study only reported the “amount” information, both metrics should be always presented. Another limitation of this study is that the metric used in this study (and other published studies from same parent project) was unique for dairy industry in Canada (mg or course dose per animal). As a result, comparing with other studies that used ‘mg/kg’ metric which is very common in poultry and swine industry is not possible (ESVAC and FAO report).
Response: We agree with your observations and added an Appendix A in the article (results in DCDbovCA /100 cow-years) and a paragraph in discussion (mg/PCU). Our objective was not to compare the quantities used in cattle with the quantities used in other species, but to propose a method for quantification of the AMU in feed (in Québec dairy farms). As you can see in the Appendix A, the overall agreement (total AMU rate, MIA usage rate) was slightly different using the unit “DCDbovCA” compared to the results obtained using the unit “kilograms”, but it did not change the interpretation of the CCC (moderate agreement for total MIA usage rate and substantial agreement for total AMU rate, between the REF method and each of the alternative methods).
Specific comments:
Line 67-68: please provide more detail explanation why authors choose the metric: ‘grams/100 cow-years’ in this study.
Response: We did not initially report the quantities using the DCDbovCA unit, because the DCDbovCA values have not been assigned for several of the antimicrobial agents identified in feed (including decoquinate, neomycin, streptomycin, benzylpenicillin, sulfamethazine, and combined trimethoprim and sulfadiazine).
We added some details in discussion (lines 430-448) to explain our choice, but we also refer now to Appendix A for results using the DCDbovCA unit.
Line 249-250: in medicine CCC value <0.7 is considered as moderate correlation only. Please consider to revise this level of agreement.
Response: We decided to keep the strength of agreement as defined by Landis and Koch (reference 18 in our article) as we already used this scale for our article “Comparison of Quantification Methods to Estimate Farm-Level Usage of Antimicrobials Other than in Medicated Feed in Dairy Farms from Québec, Canada”. We agree with you that the interpretation of the coefficient (poor, slight, fair, moderate, substantial, almost perfect agreement) varies from one author to another. This is the reason why we decided to present the value of CCC (and 95% CI) for each comparison in Table 2, not just the interpretation (the reader can use this value and have a different interpretation).
Figure 2: please add CCC value and p-value in the plot
Response: We added the CCC value and its 95% confidence interval in Figure 2, as well as the mean difference (and 95% limits of agreement) in the Bland-Altman plots.
Table 2: It seems that all of the CCC was not significant, please add one more column for p-value. Please clearly mention whether the correlation of those AAI were statistically significant or not? It is clearly that except for monensin and decoquinate, the percentage of user of other antimicrobial agents were very low (<3%), calculation of CCC for those AAI were not that helpful because it is not enough data point to obtain statistically significant.
Response: In table 2, we present each CCC value with its 95% confidence interval. For example, for monensin, when REF and VET methods are compared, a CCC value of 0.68 was found, with a 95% confidence interval from 0.57 to 0.76. We believe that it is more interesting for a reader to have the 95% confidence interval instead of the p-value. With the p-value, we could only say that the CCC is significantly different from zero.
Using the 95% CI, it is possible to say if the AMU rate estimated with one method is significantly different from the AMU rate estimated with another method (in this case, no overlap of the 95% CI would be observed).
We were able to calculate and obtain correlations for antimicrobials where the percentage of users was low because a quantity was still reported (equal to zero). The correlation could be substantial if two methods identified the same (or almost the same) non-users as well as the users (example of the comparison between REF and VET for sulfamethazine: CCC = 0.73 with 2% and 7% of users identified respectively by REF and VET).
Discussion
Authors discuss thoroughly the strength and weakness of the study and compare the finding with other published studies. However, due to the choice of metrics, the authors were unable to compare the result with other animal species: AMU in medicated feed in swine and poultry, for example. There are several publications related to AMU in medicated feed in swine and poultry recently.
Response: As mentioned previously, the main objective was to compare, not to report quantities. However, we added a paragraph in discussion with comparisons of our results (quantities using the metric mg/PCU) to other animal species – lines 443-448.